# Leiomyosarcomas of the Great Saphenous Vein: Diagnostic and Therapeutic Strategies to Prevent Unplanned Excisions and Improve Oncologic, Functional, and Psychological Outcomes

**DOI:** 10.3390/diseases13100330

**Published:** 2025-10-06

**Authors:** Luis R. Ramos Pascua, Elena Ramos García, Manuel Robustillo Rego, Violeta González Méndez, Ana Belén Enguita Valls, María I. Mora Fernández, Gabriel Rubio Valladolid, Jesús E. Vilá y Rico

**Affiliations:** 1Department of Trauma and Orthopaedics Surgery, University Hospital 12 de Octubre, Av/Córdoba, s/n, Usera, 28041 Madrid, Spain; 2School of Medicine, Complutense University, Pl. de Ramón y Cajal, s/n, Moncloa-Aravaca, 28040 Madrid, Spain; abenguita@hotmail.com (A.B.E.V.);; 3Department of Trauma and Orthopaedics Surgery, University Hospital Quirónsalud, C/Diego de Velázquez, 1, Pozuelo de Alarcón, 28223 Madrid, Spain; 4Department of Psychiatry, Hospital General Universitario Nuestra Señora del Prado, Ctra/Madrid-Extremadura, Km 114, s/n, 45600 Talavera de la Reina, Spain; 5Department of Plastic and Reconstructive Surgery, University Hospital 12 de Octubre, Av/Córdoba, s/n, Usera, 28041 Madrid, Spain; robustillorego@gmail.com; 6Department of Radiology, University Hospital 12 de Octubre, Av/Córdoba, s/n, Usera, 28041 Madrid, Spain; gonzalezmendez.violeta@gmail.com; 7Department of Pathology, University Hospital 12 de Octubre, Av/Córdoba, s/n, Usera, 28041 Madrid, Spain; 8Department of Trauma and Orthopaedic Surgery, University Hospital of Burgos, Av. Islas Baleares, 3, 09006 Burgos, Spain; 9Department of Psychiatry, University Hospital 12 de Octubre, Av/Córdoba, s/n, Usera, 28041 Madrid, Spain

**Keywords:** soft tissue sarcoma, leiomyosarcoma, great saphenous vein, excision, unplanned excision, subcutaneous, superficial, thigh, knee, ankle

## Abstract

Background/Objectives: There are very few publications on unplanned excisions of great saphenous vein leiomyosarcomas (GSV-LMS), and their impact on the prognosis of the disease is not well known. The objective of this study is to present a series of nine new leiomyosarcomas of the great saphena vein (LMS-GSV) cases, with the aim of increasing diagnostic awareness and proposing guidelines for therapeutic management. Methods: This is a prospective single-centre study of a series of nine GSV-LMS in thigh (stage IIIA AJCC), knee and proximal leg (IB, 1 II and 3 IIIA), and ankle (2 IIIB and 1 II). Eight patients were female, and the mean age was 72 years. All patients were surgically treated. Five (56%) were unplanned excisions. All these patients were reoperated on to attempt wide resection margins. In a patient, an infra-patellar amputation was performed. Another amputation was refused by another patient. Eight patients received adjuvant radiotherapy. Results: One patient died 8 years after amputation for a reason other than LMS. The patient who refused amputation has been alive, disease-free, for 28 months. The mean follow-up of surviving patients was 39 months (6–78 months). In these patients, there were no local recurrences or metastases. The mean functional outcome according to the MSTS score was 28.9 (range: 24–30). Conclusions: Unplanned excision of GSV-LMS can be prevented through clinical and imaging suspicion. Surgery and re-excision in case of inadequate previous margins and adjuvant radiotherapy lead to a good oncological and functional outcome.

## 1. Introduction

Vascular leiomyosarcoma (LMS) is a rare mesenchymal tumour that originates in the smooth muscle of the vessel wall. Most of them occur in the gastrointestinal tract, uterus, or gonadal veins, generally as sarcomas with a high grade of malignancy (grade 2 or 3). LMS affecting the venous system are extremely rare, but when they do, they are most commonly located in the inferior vena cava, followed by the renal vein, the femoral vein, and the great saphenous vein (GSV). This vein is the most frequent site of origin in the lower extremities.

The venous system of the lower limb is classified into deep (subaponeurotic) and superficial (subcutaneous) veins [1]. The deep veins, as in the upper limb, accompany the major arteries, sharing their nomenclature and anatomical relationships. The superficial veins form a subcutaneous plexus, which includes the veins of the foot (both dorsal and plantar), the small saphenous vein, and the great saphenous vein (GSV). The GSV originates as a continuation of the medial marginal vein, ascending anterior to the medial malleolus, along the medial aspect of the leg and knee, and then along the anteromedial aspect of the thigh. Approximately 3–4 cm inferior to the inguinal ligament, it penetrates the fascia and drains into the femoral vein. A thorough understanding of this anatomy is critical for the accurate diagnosis and management of related pathologies and for minimizing the risk of unexpected malignancy diagnoses and their associated oncological, functional, and economic consequences [2].

As of 2023, four literature reviews had identified a total of 48 cases reported across 45 publications [3,4,5,6]. In the latest literature review conducted in 2025, compiling cases from 1868 to 2024, a total of 56 cases in 53 reports were identified [7]. To this number, one additional unreported case involving the thigh should be added [8]. Among the 61 total papers, 57 presented a solitary case [3,4,5,6,7,8,9,10,11,12,13,14,15,16,17,18,19,20,21,22,23,24,25,26,27,28,29], and 3 presented 2 cases each [30,31,32]. The objective of this study is to present a series of nine new leiomyosarcomas of the great saphena vein (LMS-GSV) cases, with the aim of increasing diagnostic awareness and proposing guidelines for therapeutic management. To our knowledge, this is the largest series of LMS-GSV treated by a single multidisciplinary team ever published in the world. Most of the cases were unexpectedly diagnosed after surgery, upon histological workup of the specimen in non-specialized centers.

## 2. Materials and Methods

Between January 2011 and February 2025, a prospective review was conducted at our Musculoskeletal Tumor Unit, including all patients diagnosed with leiomyosarcoma (LMS) originating from the great saphenous vein (GSV) or its tributaries. A total of nine patients met the inclusion criteria. Tumor location was classified anatomically as follows: mid-thigh (*n* = 1), knee region including the proximal leg (*n* = 5), and ankle (*n* = 3). Demographic data, clinical presentation, referral pathway, imaging studies, biopsy techniques, and staging results were collected. Patient sex and age were recorded; eight patients (89%) were female, with a mean age of 72 years (range: 59–83 years). Clinical presentation in all cases consisted of a progressively enlarging subcutaneous mass, with symptom duration ranging from 2 months to 5 years. Associated symptoms such as local discomfort (*n* = 3) or pain on palpation (*n* = 1) were documented. Five patients (55.6%) were referred to our unit following unplanned excision procedures performed by general orthopedic surgeons (Cases 1, 3, and 7) or dermatologists (Cases 4 and 5), without initial suspicion of malignancy. In one case (Case 4), the tumor was misdiagnosed as dermatofibrosarcoma and treated with Mohs micrographic surgery, resulting in intralesional resection. For these patients, previous imaging—particularly magnetic resonance imaging (MRI)—histological slides, and surgical reports were reviewed to confirm diagnosis and assess resection margins. The remaining four patients (Cases 2, 6, 8, and 9) (Figure 1, Figure 2 and Figure 3) underwent standard diagnostic workup, including MRI followed by ultrasound-guided core needle biopsy. Biopsy was always performed after completion of local imaging. In all nine cases, staging studies for metastatic disease—including thoracic imaging and, when indicated, PET-CT—were negative at the time of diagnosis. Relevant imaging findings and pathological data were documented (Table 1).

The imaging findings of the cases that underwent planned management are presented in Figure 2 and Figure 3. In all of these cases, imaging suggested a malignant process, with a presumptive diagnosis of leiomyosarcoma proposed in two patients (Cases 2 and 8). In contrast, the cases that underwent unplanned excision had been previously misinterpreted as nonspecific or indicative of benign pathology (Table 1). The maximum tumor diameter, as measured on MRI, was less than 5 cm in two cases (Cases 1 and 8) and greater than 10 cm in two others (Cases 6 and 7). In the remaining five patients, tumor size ranged from 5 to 10 cm. Based on tumor size and histological grade, all cases were classified according to the AJCC staging system (Table 1). The table also details the surgical margins of the unplanned excisions in Cases 1, 3, 4, 5, and 7. Histopathological and immunohistochemical analyses were consistent across both planned and unplanned groups. These findings are summarized and illustrated in Figure 4.

All patients were surgically treated either as primary tumours or by widening the surgical margins of a previous unplanned resection according to well-established oncological principles [33,34,35,36]. In patients treated as primary cases, wide excision was performed with lateral and superficial margins of at least 2 cm from the lateral and superficial edges of the tumor. The great saphenous vein was systematically identified and ligated both proximally and distally at a minimum distance of 2 cm from the tumor edges (Figure 3D and Figure 5). The deep margin was considered oncologically adequate when the resection included the underlying fascia. In cases previously subjected to unplanned excision, re-excision was guided by preoperative MRI of the surgical site and was performed with a minimum margin of 3 cm from all aspects of the prior surgical scar, regardless of whether residual disease was apparent on imaging. Indications for adjuvant treatment are detailed in Table 2. Radiotherapy was recommended in all cases, with the exception of Case 1, due to the rapid and unfavorable progression of disease. Chemotherapy was not administered in any high-grade tumors, as the risk–benefit profile was deemed unfavorable, primarily due to the advanced age of the patients.

Case 1 underwent an infrapatellar amputation due to multiple local recurrences and the inability to achieve oncologically safe margins through a limb-sparing procedure. In Case 7, a treatment plan consisting of amputation or wide re-excision combined with ankle arthrodesis and adjuvant radiotherapy was proposed. However, the patient declined both options and opted for conservative management at another institution, without undergoing arthrodesis. Despite this, long-term follow-up continued at one of the authors’ institutions (MMF). In the remaining cases, surgical treatment consisted of wide local excision, including overlying skin (encompassing biopsy or previous surgical tracts) and deep resection to the level of the fascia. In three cases involving the knee region (Cases 3, 8, and 9), partial resection of the pes anserinus tendon insertions was required due to tumor involvement. Reconstructive procedures varied according to defect size and location. Direct wound closure was achieved in three patients (Cases 2, 5, and 6), while four cases required soft tissue coverage using a local or regional flap in combination with a thigh split-thickness skin graft (Cases 3, 4, 8, and 9). Detailed surgical techniques and reconstructions are summarized in Table 2.

We performed a descriptive analysis of the cases in the series and analyzed the response to treatment up to the present or to the patient’s death. We recorded overall surgical complications, as well as oncologic, functional, and psychological outcomes at the end of the study. Oncologic outcomes were evaluated in terms of recurrence, metastasis, and survival according to current standards. Functional outcomes were evaluated according to the MSTS scale [33]. Finally, psychological outcomes were assessed according to the HADS [37], GAD-7 [38], and QLQ-C30 [39] scales. This last assessment was carried out by the psychiatrists of our multidisciplinary team. The study was submitted for consideration and approved by the Ethics Committee of our center.

## 3. Results

The epidemiological, clinical, and diagnostic data of the cases in the series are detailed in Table 1. Surgical and adjuvant treatment data are shown in Table 2. Comparative results with other experiences are shown in Table 3.

Until the end of the study, no patient had local recurrences or metastases. Case 1 died of a disease other than LMS 8 years after amputation. The remaining 8 patients are still alive, with a mean follow-up time of 39 months (range: 6–78 months). Case 3 had neuropathic pain that required treatment with pregabalin and duloxetine prescribed by the Pain Unit. Currently, she has improved and reports tolerable pain that only requires occasional analgesics. Case 7 had pathological local metabolic uptake in a control PET-CT performed 28 months after the first intervention, which did not rule out local recurrence. For this reason, she continues to be closely followed up. The rest of the patients had no surgical or oncologic complications. Functionally, all survivors lead usual active lives and continue with the planned follow-up with a mean MSTS score of 28.9 (range: 24–30).

The psychometric scales revealed mild, reactive anxiety symptoms at the time of cancer diagnosis in most patients in our sample. However, no results compatible with the suspicion of any clinically established anxious or depressive disorder were recorded, neither at the time of diagnosis nor after 1 year of follow-up. Furthermore, all patients reported a good perceived quality of life 1 year after the oncological diagnosis. Under these conditions, and despite the possibility of being assessed by the Mental Health team, no patient in our series ended up requiring any psycho-oncological intervention.

There were no significant differences when comparing oncological and functional outcomes between planned (Cases 2, 6, 8, and 9) and unplanned excision (Cases 3, 4, 5, and 7) groups. No recurrences were recorded in either group, and the functional results according to the MSTS scale were 29.5 and 28.25, respectively. There were also no differences in the psychological results of the patients in each group.

## 4. Discussion

Unplanned excisions were first described by Giuliano and Eilber [40] and later by Noria et al. [41] to refer to tumor resections performed without preoperative diagnostic tests, mainly magnetic resonance imaging (MRI), and without the intent to achieve adequate surgical margins (negative margins) according to the Enneking system [33], where wide and radical excisions are considered negative (R0 resection) [34]. There is strong evidence to support that wide margins provide better local recurrence outcomes than intralesional or marginal margins [35]. On the other hand, the management of a patient undergoing UE is well established, with re-excision being the standard treatment after local and systemic staging of the disease. Adjuvant therapies are decided on an individual basis [36]. These concepts were applied to the cases of our series.

The incidence of UE of soft tissue sarcomas is estimated at 10–12% of cases [42], although there are historical statistics that reach 53% [2,43]. On the other hand, between 11% and 33% of all soft tissue sarcomas (STS) are located in the subcutaneous region [44]. These STS are usually smaller and more exposed than deep sarcomas to UE with inadequate resection margins [2,44]. Likewise, this eventuality seems to be more frequent in the upper limb than in the lower limb [2].

There is no evidence in the scientific literature of an association between UE and a specific histological subtype of STS. However, leiomyosarcomas (LMS) have been the most frequently published [42,44]. Dyrop et al. [44], for example, published 13 cases among 61 UE, although their location was not specified.

LMS are malignant smooth muscle tumours, which account for less than 10% of all STS [45]. They may be classified as cutaneous (confined to the dermis), subcutaneous (subdermal and suprafascial), and deep (subfascial or intramuscular) [45,46], arising from the pili-erector or surrounding sweat gland muscles of the skin or from vascular smooth muscle of the subcutis or deep soft tissues (vascular LMS), respectively. Subcutaneous LMS accounts for 1–2% of all STS with a less favorable prognosis [8,47]. Vascular LMS are quite unusual, representing less than 2% of all LMS, and affect the veins about five times more often than the arteries [11]. The inferior vena cava accounts for 36% to 60% of large-vessel LMS [20]. Retroperitoneal veins may also be sites of occurrence. The most commonly affected vessel in the lower extremity is the GSV, accounting for almost 30% of all venous LMS. It has been estimated that there is one case per million malignant tumors [4,5]; most of them develop in regions close to the knee [3,4,5,28]. Macarenco et al. [19] presented a bifocal case located in leg and ankle confined to the lumen of the vein, without disrupting the adventitia; Bibblo and Schroeder [11] described another one affecting the ankle. In our series, one case was located in the thigh, five in the knee area, and three in the ankle, which proves that any point of the GSV is susceptible to the development of an LMS. To the best of our knowledge, this series is the largest among those ever published to date [3,4,5,6,7], with practically all publications referring to a single case [3,4,5,6,7,8,9,10,11,12,13,14,15,16,17,18,19,20,21,22,23,24,25,26,27,28,29]. Wellings et al. [45] presented 30 patients with subcutaneous LMS of the lower extremity, but did not specify the affected vein, nor did they detail the cases.

LMS generally develops in endovascular to exovascular direction, although neoplastic cells may grow along the lumen of the vessels and are usually limited within that space [21]. In large vessels, growth is usually slow, with three stages being observed: nonocclusive, occlusive, and terminal stage, usually with distant metastases at this point. A pseudocapsule easily identifiable on surgical dissection is commonly observed. Histologically, they have a typical pattern of interlacing, sweeping bundles of spindle-shaped cells, with elongated occasionally torquated, blunt-ended nuclei [30,45]. Granular cell changes may be rarely found, although they have also been reported [19].

In general, the average age of presentation of the LMS-GVS is 54–60 years (range 2–85 years), with a 3:2 female-to-male ratio [26]. The lesion, although usually presenting as a nonspecific painless mobile and palpable mass, with a median size of 4.1 cm (range 2–12 cm) [18], can also present through a number of other symptoms and signs. In this respect, oedema, lymphadenopathy, or simulated thrombosis, may also be observed and lead to misdiagnosis [6,29]. Clinical suspicion depends on the stage of disease progression. In nonocclusive stage, the lesion is usually asymptomatic, and the diagnosis would be incidental. In the occlusive form, the disease may appear as a phlebitis and/or an unilateral oedema on physical examination, although the patient often does not manifest any symptoms because of sufficient collateralisation. At other times, it may be mistaken for thrombophlebitis and lymphedema, without being either. At this stage, the lesion is usually palpable, of variable consistency, and can be laterally mobilised at any point along the course of the vein [11]. It may also appear adherent to deep planes and be painful on palpation (tender). In the final metastatic stage, the diagnosis would be easier to make [11]. Thus, a high index of clinical suspicion is very important for an early diagnosis and makes it necessary to complete the study by means of imaging tests. In a UE series in which 50% were superficial small tumours, 60.9% of them had not received any image diagnosis before surgery [44]. In our series, the most outstanding epidemiological data were the predominance of women (89%) and the lack of diagnostic suspicion in more than half of the cases (56%). The higher incidence of LMS-GSV in women could be the subject of study for future research, as there may be other unknown sex-related factors involved. Regarding the influence of gender on disease prognosis, a possible protective effect of estrogen in hormonally active women has been suggested, although this hypothesis has been challenged by some authors in whose series all patients who died due to superficial leiomyosarcomas were female, with a mean age of 48 years [45].

The diagnosis of an LMS-GSV begins with the knowledge of the disease and the path of the vein in the subcutaneous cellular tissue of the lower limb. Basic training in tumor pathology and experience are fundamental elements to avoid diagnostic errors. Regarding this experience, Weber et al. reported that most orthopedic surgeons are likely to see a patient with a benign or malignant musculoskeletal tumor at some point in their career [48]. Moreover, it is surprising that nearly 50% of deep soft tissue sarcomas resections are performed by non-oncology-designated surgeons and that approximately 17% are performed by practitioners who complete an average of one to two of these procedures per year [49]. The five unplanned excisions in our series were due to a lack of clinical suspicion of the disease.

Clinical suspicion makes it necessary to perform complementary imaging tests, among which an ultrasound scan is recommended and an MRI is mandatory. Doppler ultrasound can show tissue heterogeneity, bulging, vein deviation, peritumoral vascularization, and localized flow rates during the nonocclusive stage. It can also help rule out extrinsic compression, mural thrombus, or post-thrombotic syndrome [3,9,11]. These and other benign pathologies can be mistaken for an LMS-GSV [6,22,30]. MRI is necessary to determine local extension and differentiation from other soft tissue sarcomas, although the findings are nonspecific. Transverse sections may show low signal intensity on T1-weighted sequences, high signal intensity on T2-weighted sequences, and enhanced intensity on T1-weighted images with gadolinium injection. MRI angiography could be of interest, but does not seem necessary. Once the diagnosis has been confirmed by biopsy, the extent of the study of the disease must be completed. Metastases may develop early from large-vessel LMS, and in at least 10% of the cases, they are already present at initial diagnosis [21].

Regarding the treatment of the LMS-GVS, currently the best option seems to be a wide excision of the tumour, combined with adjuvant radiotherapy depending on the tumour grading. The re-establishment of vascular continuity of the saphenous vein is not mandatory [3]. Chemotherapy is reserved for cases where metastases occur [28], although some regimens (doxorubicin and ifosfamide, trabectedin, dacarbazine, pazopanib, or gemcitabine and Taxotere) may result in long survival periods and should be taken into consideration in selected cases. However, the indications, effectiveness, and protocols of adjuvant radiochemotherapy remain unclear [29]. In the event of an unanticipated diagnosis of malignancy with intralesional or marginal margins, the excision should be enlarged with appropriate skin coverage and adjuvant treatment if considered [18]. In our series, we tried to achieve a wide resection margin in all non-previously-treated cases and to widen it to at least 3 cm away from the previous scar in UE cases. In these and in those with a high degree of malignancy, we complemented surgery with radiotherapy, taking into account the fact that the saphenous vein is superficial and the subcutaneous space is not compartmentalized. To date, we have achieved good oncologic results in all cases. The neuropathic pain reported by Case 3 was attributed to a surgical complication directly related to the resection of terminal branches of the saphenous nerve and the use of a graft for soft tissue coverage. This eventuality, which is sometimes challenging to treat, should be taken into consideration due to the proximity of the nerve to the GSV.

After treatment, patients should be regularly monitored for recurrence, which most commonly occurs 2–3 years later [3]. It is therefore suggested to carry out routine chest radiographs and site-specific MRI every 6 months during the first 5 years (and, subsequently, on an annual basis) after a first baseline 3–4 months after surgery.

In general, the prognosis for primary venous LMS is poor, with a median survival period of only 30–40 months [20]. However, the prognosis for superficial LMS seems to be rather more favorable, particularly that of dermal tumors [19,29,39]. Five-year survival is estimated at 90% and 80% in low-grade and high-grade, respectively [4,18]. Table 3 shows a comparative summary between the results of treatment of individual cases published and ours, showing a good oncologic, functional, and psychological outcome regardless of tumor grade and the time and type of diagnosis. However, most of the publications to date are limited to sporadic cases with a short follow-up time.

Although UE per se does not seem to be associated with a shortened survival of patients and despite the fact that the issue is controversial [45], the prognosis depends on the degree of malignancy and the presence or not of tumor remnants at re-excision [50]. High-grade STS have a worse prognosis than low-grade ones, as do deep-seated versus superficial cases. However, more than 60% of superficial cases may have a high degree of malignancy [44]. In the LMS-GSV group, the percentage of UE is estimated to be higher than half of the cases (72% in the Wellings et al. series [45] and 55.6% in ours).

Regarding the presence of residual disease on the tumor bed during re-excision, it has been shown to worsen oncologic outcomes [43]. In this context, Larios et al. [50] suggested the possibility that some histological subtypes may be at higher risk for residual disease on re-excision. In another recent review on the subject, Grimer et al. [43] considered leiomyosarcomas, together with liposarcomas, to be one of the tumours with the lowest risk of leaving residual disease after UE. At the opposite end of the spectrum would be dermatofibrosarcoma protuberans, malignant peripheral nerve sheath tumor, and myxofibrosarcoma, followed by synovial sarcoma and undifferentiated pleomorphic sarcoma [43]. Nevertheless, economic and emotional considerations aside, the UE of an LMS-GVS should not be associated with a significant worsening of the disease prognosis, even in high-grade cases. This controversial fact seems to be deduced from our results.

In any case, it is important to remember some fundamental aspects in the management of soft tissue sarcomas, starting with the needed suspicion to reduce unexpected diagnoses of malignancy. As established in the guidelines, any soft tissue mass deeper than the fascia and larger than 5 cm in diameter, whether painful or not, especially if it grows or reappears after resection, is a sarcoma until proven otherwise [51]. The same axiom should apply to subcutaneous tumours, bearing in mind that one third of soft tissue sarcomas are superficial and that their size is usually smaller than that of deep SPBs [43]. Cases should be centralized in referral centers, the pathology of unplanned excisions should be reviewed, and therapeutic decisions should be made by multidisciplinary teams. In this sense, specific educational programs for all surgical specialties are very important to minimize the differences in criteria between them [52].

Our series is not free of limitations. The main one is the size of the sample. Another potential limitation of this study is that the diagnosis of LMS originating from the great saphenous vein (GSV) was inferred based on the anatomical location of the lesions, as determined by imaging studies, which showed exclusive involvement along the GSV trajectory. In Case 1, it was not possible to confirm whether the tumor originated from the internal dorsal vein or its tributaries on the medial side of the dorsal venous arch of the foot. Although less likely, a similar consideration could apply to the more proximal tumors, given that the GSV receives venous drainage from several tributaries along its course, including accessory saphenous veins and communicating veins, which connect the superficial and deep venous systems. The third limitation is that we did not classify our cases into histologic subtypes of LMS, but rather considered them collectively as subcutaneous leiomyosarcomas of varying malignant grade. Although LMS subtypes share uniform histologic features, their clinical behavior and characteristics differ substantially across anatomic sites (uterus, retroperitoneum, vena cava, subcutaneous or cutaneous tissues, etc.), and molecular studies appear necessary to assess the efficacy of novel immunotherapeutic approaches [53]. The last limitation may be the fact that one patient was treated at another centre, although we learned from their subsequent evolution. We believe that all these limitations are offset by the strengths of the study, beginning with its prospective design. Moreover, this is the largest series ever published; the final treatment was performed by the same surgeon (LRRP) and multidisciplinary team in eight of the nine cases; the patient’s follow-up is longer than that of most cases published to date. In this regard, the average follow-up period for our series was 39 months, with six patients having more than 3 years of follow-up and three of them having more than 6 years. In any case, an active surveillance approach is essential for the early detection of local recurrences [54], especially in high-grade LMS.

## 5. Conclusions

Continuous education on unplanned diagnosis of malignancy in subcutaneous soft tissue lesions prevention standards is essential to avoid unplanned excisions. A thorough understanding of the histology and topographic anatomy of the great saphenous vein can facilitate early diagnosis and appropriate management of LMS at this site. Surgical excision—with re-excision if previous margins were inadequate—combined with adjuvant radiotherapy, can result in favorable oncological, psychological, and functional outcomes.

## Figures and Tables

**Figure 1 diseases-13-00330-f001:**
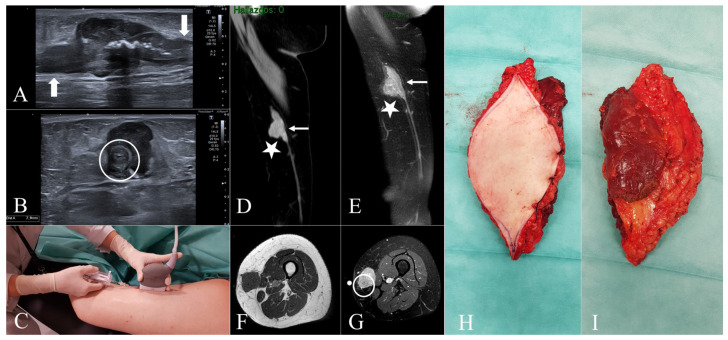
Case 2. (**A**,**B**) = Ultrasound images show a well-defined solid subcutaneous soft tissue mass. The tubular structures on both ends (**A**) correspond with the great saphenous vein (white arrows). Note the mural thickening of the great saphenous vein (**B**), which is encased by the mass (white circle). (**C**) = Ultrasound-guided core needle biopsy. The biopsy should be planned so that the biopsy tract can be safely removed at the time of definitive surgery. (**D**,**E**) = Sagittal post-contrast CT and sagittal post-contrast T1W fat-suppressed images reveal mural thickening of a segment of the great saphenous vein (white arrows) and a polylobate soft tissue saphenous-dependent mass (white stars). (**F**) = Axial T1W images exhibit a well-circumscribed, isointense to muscle, and subcutaneous soft tissue mass. (**G**) = DP fat-suppressed axial image shows the hyperintensity of the mass and its relationship with the great saphenous vein (white circle), well correlated with transverse grayscale US (**B**). (**H**) = Superficial side of the specimen, including a large segment of skin. (**I**) = Deep side of the specimen, including the fascia and a muscular cuff to ensure a wide resection margin. Note the inclination of the biopsy needle so as not to cross the fascia and avoid contamination of healthy deep planes of the thigh. Also note the peripheral sealing of the specimen by means of stitches between the skin and the fascia (**H**).

**Figure 2 diseases-13-00330-f002:**
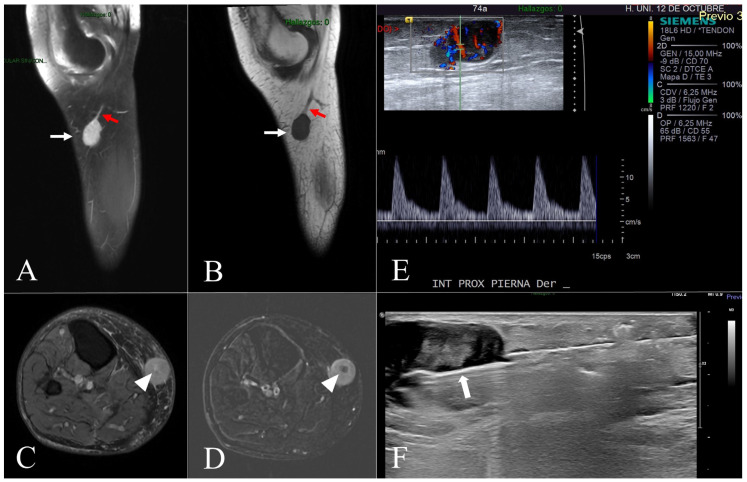
Case 8. (**A**,**B**) = Sagittal DP fat-suppressed (**A**) and sagittal T1-weighted (**B**) MR images show a subcutaneous well-defined and homogeneous lesion (white arrows) in continuity with the saphenous vein (red arrows). (**C**,**D**) = Axial DP fat-suppressed (**C**) and digital subtraction post-processing images show homogeneous enhancement with central low attenuation representing necrosis (white arrowheads). (**E**) = A longitudinal ultrasound view of the right leg demonstrates a well-defined lesion located in the subcutaneous fat layer with patent internal vascularity. (**F**) = A longitudinal ultrasound shows the biopsy needle (white arrow), which has been advanced into the lesion under US guidance while avoiding crossing the fascia of the leg.

**Figure 3 diseases-13-00330-f003:**
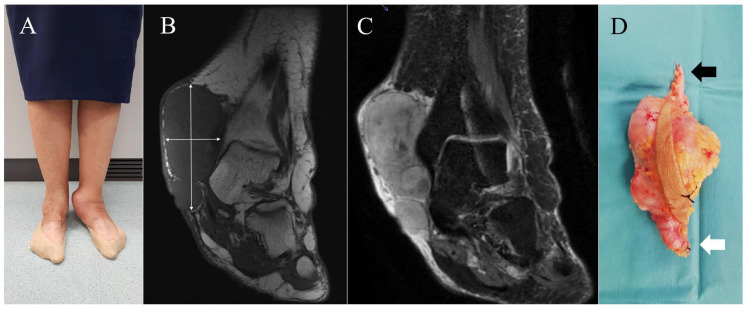
Case 6. (**A**) = Clinical appearance of a woman with a large tumor on the inner aspect of her left ankle. (**B**,**C**) = Coronal T1-weighted and coronal T2-weighted, fat-suppressed MR images exhibit a giant subcutaneous lobulated nonspecific mass (blue arrows). In this case, continuity with vascular structures was not established, probably due to its large size. The lesion is in contact with tibial epiphysis, but no associated osseous changes were demonstrated. (**D**) = Specimen with marginal margin of resection. The skin island that includes the biopsy tract is identified; the ends of the great saphenous vein are indicated: proximal end (black arrow) and distal end (white arrow).

**Figure 4 diseases-13-00330-f004:**
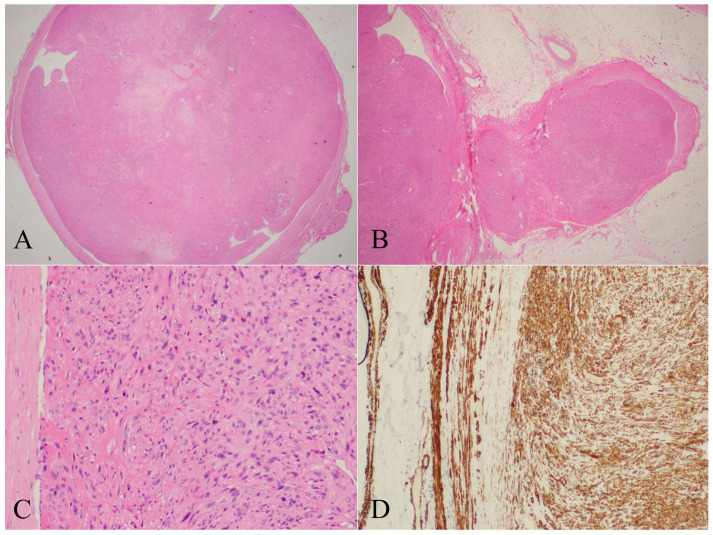
Case 5. (**A**) = H-E 2X. Mesenchymal proliferation that occupies almost all the vascular lumen. (**B**) = H-E 4X. Cellularity of fusiform morphology, with blunt-edged nuclei, with cytological atypia. (**C**) = H-E 20X. Tumor cellularity at higher magnification. Peripheral area of the vascular wall. (**D**) = IHC 10X. Desmina. Diffuse positivity of neoplastic cellularity, with positive control of the vascular wall.

**Figure 5 diseases-13-00330-f005:**
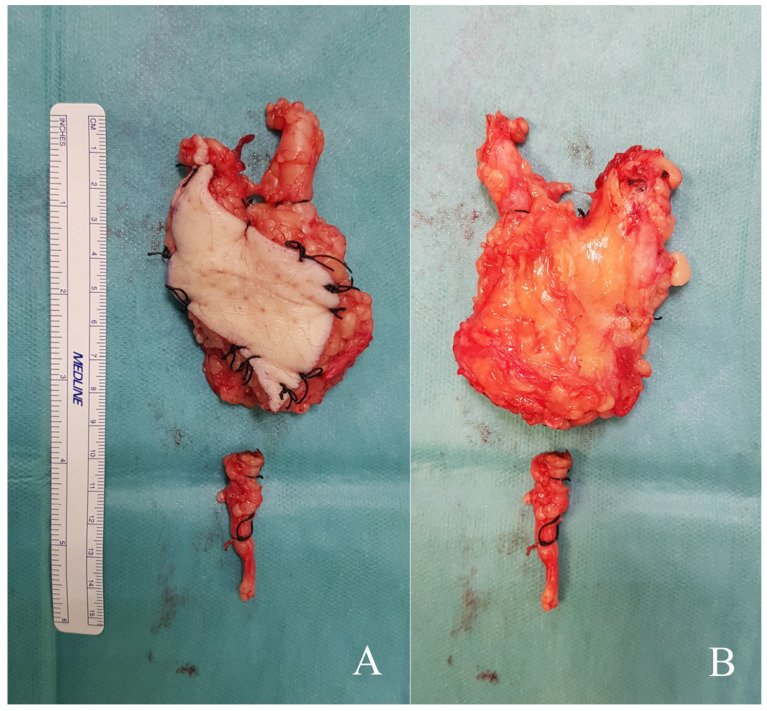
Superficial (**A**) and deep (**B**) side of the specimen of Case 5, including the fascia and a muscular cuff to ensure a wide resection margin. Note the ligated ends of the great saphenous vein at the proximal and distal aspects of the specimen, with the distal segment of the vein separated for better visualization.

**Table 1 diseases-13-00330-t001:** Summary of epidemiological, clinical, and diagnostic data of the patients in the series.

Case	Age	Gender	Notable Antecedents	Location	Side	Diagnosis (MRI)	UE	Stage (AJCC)
1	80	F	Fibromatosis (foot)	Ankle	R	NS	Yes (IM) ^a^	II
2	71	F	-	Thigh	L	L, LMS, HE	No	IIIA
3	59	F	-	Knee	L	Schwannoma	Yes (UM)	IIIA
4	60	F	Contralateral infrapatellar amputation	Knee	L	DMFS	Yes (IM)	IIIA
5	75	M	-	Knee	L	NS	Yes (UM) ^b^	IB
6	83	F	-	Ankle	L	Suggests sarcoma	No	IIIB
7	69	F	-	Ankle	R	Ganglion	Yes (IM)	IIIB
8	74	F	SMM	Proximal leg	R	Suggests LMS	No	II
9	74	F	Diabetes. Stroke	Proximal leg	R	Suggests sarcoma	No	IIIA

NS = not specified. SMM = smoldering multiple myeloma. R = right. L = left. L = leiomioma. LMS = leiomyosarcoma. HE = haemangioendothelioma. DMFS = dermatofibrosarcoma. UE = unplanned excision. IM = intralesional margin. UM = unknown margin. AJCC = American Joint Committee on Cancer. ^a^ Rapid local recurrence. ^b^ Provide photograph.

**Table 2 diseases-13-00330-t002:** Summary of treatment and outcomes of patients in the series.

Case	Resection (Margin)	Residual Tumor	Reconstruction	Postoperative Hospital Stay (Days)	Adjuvant RT/QT	Follow-Up (Months)	Oncologic Result (LC/MTS)	Functional Result (MSTS)
1	Amputation (Wide)	Yes	Amputation leg	17	No/No	96	No/No (exitus)	-
2	Wide	No	Direct closure	4	Yes/No	78	No/No	30
3	Re-excision (margin widening)	No	Superficial femoral artery perforator flap + FSAG	27	Yes/No	72	No/No	24
4	Re-excision (margin widening)	Yes	IGRF + FSAG	7	Yes/No	45	No/No	30
5	Re-excision (margin widening)	Yes	Direct closure	1	Yes/No	36	No/No	30
6	Marginal	No	Direct closure	1	Yes/No	34	No/No	30
7	Marginal?	No	Margin widening? ^a^	?	Yes ^b^/No	28	No? ^c^/No	29
8	Wide	No	IGRF + FSAG	8	Yes/No	12	No/No	29
9	Wide	No	IGRF + FSAG	20 ^d^	Yes/No	6	No/No	29

RT = radiotherapy. QT = chemotherapy. LC = local recurrence. MTS = metastases. IGRF = internal gastrocnemius rotational flap. FSAG = free skin autograft. MSTS = Musculoskeletal Tumor Society. ^a^ Refuses amputation and fusion ankle. ^b^ Intraoperative and postoperative radiotherapy (4 days). ^c^ Pathological metabolic local uptake in PET-CT scan performed 28 months after the first intervention, which did not rule out local recurrence. ^d^ Discharge delayed for social reasons.

**Table 3 diseases-13-00330-t003:** Comparative summary with the results of treatment of great saphenous vein leiomyosarcomas in the scientific literature of the last 20 years.

Authors	n/Grade	Gender/ Age (Years)	Location	Treatment	Follow-Up (Months)	Complications	Oncological Results	Functional Results	Psychological Results
Gross and Horton, 1975 [15]	1/High	M/46	Thigh	Simple excision	45	No	Thyroid (thyroidectomy and RT) and subcutaneous (excision biopsy) metastases (3 years). Alive	NS	NS
Berlin et al., 1984 [10]	1/High	M/60	Thigh (groin)	Extended hip joint disarticulation	1	Thrombosis and fatal pulmonary embolism	Lung and liver metastases	NS	NS
Song et al., 1991 [24]	1/High	F/54	Thigh	Simple excision	NS	NS	NS	NS	NS
Dzsinich et al., 1992 [32]	1/NS	F/70	NS	Complete excision	204	No	Alive	NS	NS
	1/NS	F/54	NS	Complete excision	9	No	Died	NS	NS
Saglik et al., 1992 [23]	1/High	F/61	Thigh	Complete excision	72	No	Local recurrence (excision + RT) and lung metastases (CHT). Died	NS	NS
Stallard et al., 1992 [25]	1/High	F/64	Groin	NS	NS	NS	NS	NS	NS
Reix et al., 1998 [21]	1/NS	M/64	NS	Complete excision + CHT	72	No	Skin, lung, and brain metastases (14 months). Died	Good	NS
Le Minh et al., 2004 [30]	1/High	F/52	Thigh (groin)	WE + RT + CHT	12	Limphoedema and local pigmentation of the skin	FD	Good	NS
	1/I	M/66	Thigh	Surgery (NS) + RT	6	No	FD	Good	NS
Van Marle et al., 2004 [5]	1/II	F/85	Groin	WE	2	Wound infection	FD	Good	NS
Zhang and Wang, 2006 [28]	1/NS	F/59	Thigh	Surgery (NS) + RT	10	No	FD	NS	NS
El Khoury et al., 2006 [13]	1/I	M/60	Thigh	WE	6	No	Alive	Good	NS
Mammano et al., 2008 [20]	1/High	M/48	Groin	WE ^c^	30	Thrombosis and proximal stenosis of the prosthesis. Lymphoedema	Lung metastases and death	NS	NS
Bibbo and Schroeder, 2011 [11]	1/I	F/66	Ankle	ME + RT ^a^	12	*Clostridium difficile* colitis and wound healing issues	FD	Good	NS
Røpcke et al., 2012 [22]	1/?	F/63	Thigh	Complete excision	NS	NS	NS	NS	NS
Werbrouck et al., 2013 [27]	1/High	M/57	Groin	Complete excision	NS	NS	NS	NS	NS
Fremed et al., 2014 [14]	1/High	M/59	Thigh	Primary excision + vein reconstruction	6	Lymphocutaneous fistula	NS	NS	NS
Lin et al., 2016 [18]	1/I	M/75	Leg	WE	6	No	FD	NS	NS
Cangiano et al., 2017 [4]	1/I	F/65	Thigh	WE	10	No	FD	NS	NS
Macarenco et al., 2018 [19]	1/II	M/57	Leg and ankle	WE ^b^ + RT	39	No	FD	Good	NS
Naouli et al., 2019 [3]	1/III	M/45	Thigh	WE	6	No	FD	NS	NS
Güner et al., 2020 [16]	1/High	M/37	Leg and ankle	WE + CHT	36	No	Local recurrence and cardiac metastases (36 months)	NS	NS
Tresgallo-Parés et al., 2021 [26]	1/High	F/67	Thigh	WE + RT	24	No	FD	Independent	NS
Alkhaled et al., 2022 [9]	1/NS	F/49	Thigh	WE	NS	NS	NS	NS	NS
Dziekiewicz et al., 2022 [12]	1/NS	F/61	Leg	Surgery (NS)	12	NS	NS	NS	NS
Atieh et al., 2023 [6]	1/NS	F/63	Groin	ILE	NS	NS	NS	NS	NS
Liu et al., 2025 [7]	1/High	M/37	Leg	WE	13	No	Liver and lung metastases (13 months). Alive	NS	NS
Ramos et al., 2025 (this research)	9/1I-2II-6III	8F-1M/72 ^d^ (59–83)	1 Thigh. 5 Knee. 3 Ankle	See Table 2	45 ^e^	No	1 local recurrence	28.9 MSTS ^f^	No anxious or depressive disorder ^f^

n = number of cases. M = male. F = female. WE = wide excision. ILE = intralesional excision. ME = marginal excision. RT = radiotherapy. CHT = chemotherapy. FD = free of disease. NS = not specified. MSTS = Musculoskeletal Tumor Society score. ^a^ After marginal excision and radiotherapy, a wide excision and skin grafting was performed. ^b^ Reconstruction was done with rotation of fascial muscle local grafts and covered with free skin graft. ^c^ After wide excision, femoral vein was replaced with a polytetrafluoroethylene (PTFE) prosthesis. ^d^ Mean age. ^e^ Mean follow-up. ^f^ Excluding the amputee patient (Case 1).

## Data Availability

The datasets used and/or analyzed during the current study are available from the corresponding author on reasonable request.

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
