# Peer review of "Leiomyosarcomas of the Great Saphenous Vein: Diagnostic and Therapeutic Strategies to Prevent Unplanned Excisions and Improve Oncologic, Functional, and Psychological Outcomes"

_diseases, 2025, doi:10.3390/diseases13100330_

Round 1

Reviewer 1 Report

Comments and Suggestions for Authors

Thank you to give me the possibility to review this paper on diagnostic management of leiomiosarcomas of the great saphenous vein.

Abstract is clear and exhaustive.

Introduction. Personally I would delete the sentence "most of them occur.....the large veins and arteries".

I would add gonadal veins in the sentence "they are most located....and the great saphenous vein(GSV)"

The description of the anatomy is a little bit too long and should be synthesized.

Matherial and methods.Authors should clarify why they did not consider neo or adjuvant chemotherapy or neoadjuvant radio+ chemo at least in the six cases of G3 and why did you employed Radio also in cases G1 and G2

In table 1 I prefer to use as stage system the G classification that is most commonly used (you can add another column).

Discussion and Conclusion. No particular observations

Comments on the Quality of English Language

Ehglish is quite good, could be improved.

Author Response

Abstract is clear and exhaustive.

Introduction. Personally I would delete the sentence "most of them occur.....the large veins and arteries".

Line 58: Corrected

I would add gonadal veins in the sentence "they are most located....and the great saphenous vein(GSV)"

Line 58: Added

The description of the anatomy is a little bit too long and should be synthesized.

Lines 63-74: It has been summarized

Matherial and methods.Authors should clarify why they did not consider neo or adjuvant chemotherapy or neoadjuvant radio+ chemo at least in the six cases of G3 and why did you employed Radio also in cases G1 and G2

We have included the following paragraph in the Materials and Methods section to clarify this point:

Lines 170-174: Indications for adjuvant treatment are detailed in Table 2. Radiotherapy was recommended in all cases, with the exception of case 1, due to the rapid and unfavorable progression of disease. Chemotherapy was not administered in any high-grade tumors, as the risk–benefit profile was deemed unfavorable, primarily due to the advanced age of the patients. 

In the Discussion section, we also explain the indication for radiotherapy and chemotherapy in LMS-GSV:

Lines 360-363 (no changes): ...Chemotherapy is reserved for cases where metastases occur [11], although some regimens (Doxorubicin and Ifosfamide, Trabectedin, Dacarbazine, Pazopanib, or Gemcitabine and Taxotere) may result in long survival periods and should be taken into consideration in selected cases.

Lines 367-370 (no changes): ...In these and in those with a high degree of malignancy we complemented surgery with radiotherapy, taking into account the fact that the saphenous vein is superficial and the subcutaneous space is not compartmentalized.

In table 1 I prefer to use as stage system the G classification that is most commonly used (you can add another column).

We have added a column with the histological grade of the cases in Table 1. Nevertheless, the AJCC classification column has been retained due to its value with respect to the size of the lesion.

Discussion and Conclusion. No particular observations

Comments on the Quality of English Language

Ehglish is quite good, could be improved.

In accordance with the reviewer’s suggestions, we have improved the writing and English translation of the manuscript. Specifically, we have revised several paragraphs within the Abstract, Introduction, Material and Methods, and Conclusion sections, while ensuring that the content remains consistent with the original version of the manuscript.

All changes suggested by the reviewers -many of which are summarised and confirmed in the tables and figures of the manuscript- have been taken into account. We sincerely appreciate the comments, which we believe enrich our paper.

Reviewer 2 Report

Comments and Suggestions for Authors

There are some comments.

It would be better to describe the pathological features (e.g., tumor size, histologic grade, and resection margin involvement) and radiologic features (e.g., MRI) of nine cases in detail.

For cases involving leiomyosarcoma in the ankle, it would be helpful to explain how to confirm that the tumor originates from the great saphenous vein.

Comparing outcomes between planned and unplanned excision groups would be beneficial, particularly in terms of MSTS scores and recurrence rates.

A detailed explanation of radiotherapy and chemotherapy should be provided.

Please write the abbreviations below in Table 2.

Please confirm the term "greater saphenous vein" vs. "great saphenous vein".

Comments on the Quality of English Language

Please check the grammar and spelling of the English text.

For example,   epiphyfisis->epiphysis
                        Each autor ->Each author

Author Response

It would be better to describe the pathological features (e.g., tumor size, histologic grade, and resection margin involvement) and radiologic features (e.g., MRI) of nine cases in detail.

A new paragraph has been added to the Materials and Methods section detailing the imaging and histological findings, which are summarised in Table 1:

Lines 149-159: The imaging findings of the cases that underwent planned management are presented in Figures 2 and 3. In all of these cases, imaging suggested a malignant process, with a presumptive diagnosis of leiomyosarcoma proposed in two patients (cases 2 and 8). In contrast, the cases that underwent unplanned excision had been previously misinterpreted as nonspecific or indicative of benign pathology (Table 1). The maximum tumor diameter, as measured on MRI, was less than 5 cm in two cases (cases 1 and 8), and greater than 10 cm in two others (cases 6 and 7). In the remaining five patients, tumor size ranged from 5 to 10 cm. Based on tumor size and histological grade, all cases were classified according to the AJCC staging system (Table 1). The table also details the surgical margins of the unplanned excisions in cases 1, 3, 4, 5, and 7. Histopathological and immunohistochemical analyses were consistent across both planned and unplanned groups. These findings are summarized and illustrated in Figure 5.

For cases involving leiomyosarcoma in the ankle, it would be helpful to explain how to confirm that the tumor originates from the great saphenous vein.

We appreciate this interesting and accurate observation because, indeed, in one of the ankle cases, it was impossible to determine whether the tumor origin was the great saphenous vein or one of its tributary veins. We have therefore included this appreciation in the paragraph on the limitations and strengths of the study, which is located in the Discussion section:

Lines 419-428: Another potential limitation of this study is that the diagnosis of LMS originating from the great saphenous vein (GSV) was inferred based on the anatomical location of the lesions, as determined by imaging studies, which showed exclusive involvement along the GSV trajectory. In case 1, it was not possible to confirm whether the tumor originated from the internal dorsal vein or its tributaries on the medial side of the dorsal venous arch of the foot. Although less likely, a similar consideration could apply to the more proximal tumors, given that the GSV receives venous drainage from several tributaries along its course, including accessory saphenous veins and communicating veins, which connect the superficial and deep venous systems.

Comparing outcomes between planned and unplanned excision groups would be beneficial, particularly in terms of MSTS scores and recurrence rates.

This is a very good suggestion. A final paragraph has been added to the Results section:

Lines 252-256: There were no significant differences when comparing oncological and functional outcomes between planned (cases 2, 6, 8, and 9) and unplanned excision (cases 3, 4, 5, and 7) groups. No recurrences were recorded in either group, and the functional results according to the MSTS scale were 29.5 and 28.25, respectively. There were also no differences in the psychological results of the patients in each group.

A detailed explanation of radiotherapy and chemotherapy should be provided.

In the Materials and Methods section, we have added the justification for complementary therapies to surgery in the series:

Lines 170-174: Indications for adjuvant treatment are detailed in Table 2. Radiotherapy was recommended in all cases, with the exception of case 1, due to the rapid and unfavorable progression of disease. Chemotherapy was not administered in any high-grade tumors, as the risk–benefit profile was deemed unfavorable, primarily due to the advanced age of the patients. 

In the Discussion section, we also explain the indication for radiotherapy and chemotherapy in LMS-GSV:

Lines 360-363 (no changes): ...Chemotherapy is reserved for cases where metastases occur [11], although some regimens (Doxorubicin and Ifosfamide, Trabectedin, Dacarbazine, Pazopanib, or Gemcitabine and Taxotere) may result in long survival periods and should be taken into consideration in selected cases.

Lines 367-370 (no changes): ...In these and in those with a high degree of malignancy we complemented surgery with radiotherapy, taking into account the fact that the saphenous vein is superficial and the subcutaneous space is not compartmentalized.

Please write the abbreviations below in Table 2.

Abbreviations have been written below Table 2:

RT = Radiotherapy. QT = Chemotherapy. LC = Local recurrence. MTS = Metastases. IGRF = Internal Gastrocnemius Rotacional Flap. FSAG = Free Skin AutoGraft. MSTS = MusculoSkeletal Tumor Society.

Please confirm the term "greater saphenous vein" vs. "great saphenous vein".

Both terms are accepted in scientific literature. However, we have unified them as ‘great saphenous vein’ in our manuscript.

Comments on the Quality of English Language

Please check the grammar and spelling of the English text.

For example,   epiphyfisis->epiphysis
                  Each autor ->Each author

We apologize for our mistakes with the English translation. The terms have been corrected throughout the manuscript. In addition, minor changes have been made on the quality of English language of the manuscript in an attempt to improve its readability. More particularly, we have revised several paragraphs within the Abstract, Introduction, Material and Methods, and Conclusion sections, while ensuring that the content remains consistent with the original version of the manuscript.

All changes suggested by the reviewers -many of which are summarised and confirmed in the tables and figures of the manuscript- have been taken into account. We sincerely appreciate the comments, which we believe enrich our paper.

Reviewer 3 Report

Comments and Suggestions for Authors

In a retrospective study, the authors elaborated on the oncological outcome of patients operated for leiomyosarcomas in the great saphenous vein. The paper tries to address the question regarding the outcome of treatment for this particular diagnosis, seen through the perspective of their unique anatomical location. This is the main novelty regarding the previous studies on leiomyosarcomas of the soft-tissues of the extremities.  I am afraid this does not suffice since the diagnosis and treatment of leiomyosarcomas  is well known, and the results in this specific location, in this small cohort (including only 9 patients), are similar to the known data for this disease without adding any significant new knowledge.  Otherwise, the methodology is correct, mainly being a descriptive study and the paper is well-written, and I have no objections on the style of the text and figures,  the references are appropriate. Overall, I believe the interest to readers is low. 

Author Response

In a retrospective study, the authors elaborated on the oncological outcome of patients operated for leiomyosarcomas in the great saphenous vein. The paper tries to address the question regarding the outcome of treatment for this particular diagnosis, seen through the perspective of their unique anatomical location. This is the main novelty regarding the previous studies on leiomyosarcomas of the soft-tissues of the extremities.  I am afraid this does not suffice since the diagnosis and treatment of leiomyosarcomas  is well known, and the results in this specific location, in this small cohort (including only 9 patients), are similar to the known data for this disease without adding any significant new knowledge.  Otherwise, the methodology is correct, mainly being a descriptive study and the paper is well-written, and I have no objections on the style of the text and figures,  the references are appropriate. Overall, I believe the interest to readers is low. 

Thank you very much for your comments. Certainly, although inadvertent resection of soft tissue sarcomas is a well-documented concern in the literature, it is striking that reports focusing specifically on superficial leiomyosarcmas arising from the great saphenous vein remain scarce. With regard to sample size, to our knowledge this is the largest series of LMS-GSV ever published in the scientific literature. Furthermore, our study design is prospective rather than retrospective. For all these reasons, we believe that the topic is relevant and worthy of publication.

Minor changes have been made on the quality of English language of the manuscript in an attempt to improve its readability. More particularly, we have revised several paragraphs within the Abstract, Introduction, Material and Methods, and Conclusion sections, while ensuring that the content remains consistent with the original version of the manuscript.

Minor changes have been made on the quality of English language of the manuscript in an attempt to improve its readability. More particularly, we have revised several paragraphs within the Abstract, Introduction, Material and Methods, and Conclusion sections, while ensuring that the content remains consistent with the original version of the manuscript.

Reviewer 4 Report

Comments and Suggestions for Authors

Ramos Pascua et al. conducted a study on leiomyosarcomas of the great saphenous vein, focusing on diagnostic management to avoid unplanned excisions and evaluating the oncological, functional, and psychological treatment results.

Despite the undeniable interest of the study, it is suggested to consider the following recommendations to enrich the manuscript:

Female Predominance: Given that the authors identify a significant predominance of the disease in women, it is recommended to extend the discussion to hypothesize about the possible reasons for this marked preference.

Value of the Follow-up Period: Although the authors describe the duration of the follow-up, it would be valuable for them to delve deeper into the discussion about the importance of this prolonged period for the validity and relevance of their long-term findings.

Diagnostic Markers: It would be beneficial to describe possible diagnostic markers that help reduce unplanned excisions, thereby reinforcing the clinical impact of the study.

In-depth Discussion of Neuropathic Pain: Although the authors mention the presence and management of neuropathic pain, it is suggested to expand this discussion. They could address how neuropathic pain affects the patient's quality of life, detail management strategies, and highlight the need for future research on this complication in this type of tumor.

Author Response

Ramos Pascua et al. conducted a study on leiomyosarcomas of the great saphenous vein, focusing on diagnostic management to avoid unplanned excisions and evaluating the oncological, functional, and psychological treatment results.

Despite the undeniable interest of the study, it is suggested to consider the following recommendations to enrich the manuscript:

Female Predominance: Given that the authors identify a significant predominance of the disease in women, it is recommended to extend the discussion to hypothesize about the possible reasons for this marked preference.

Indeed, in our series, the proportion of women was notably higher than that of men (8:1). Although this fact does not appear to be attributable to chance, we are unable to propose any hypotheses that could explain this finding. Nevertheless, this observation is further addressed in the Discussion section, where we suggest the need for increased clinical awareness and recommend further research to explore potential sex-related factors associated with this pathology. Moreover, we have added and modified the following paragraph:

Lines 326-332: The higher incidence of LMS-GSV in women could be the subject of study for future research, as there may be other unknown sex-related factors involved. Regarding the influence of gender on disease prognosis, a possible protective effect of strogen in hormonally active women has been suggested, although this hypothesis has been challenged by some authors in whose series all patients who died due to superficial leiomyosarcomas were female with a mean age of 48 years [32].

Value of the Follow-up Period: Although the authors describe the duration of the follow-up, it would be valuable for them to delve deeper into the discussion about the importance of this prolonged period for the validity and relevance of their long-term findings.

We completely agree. This is why we have placed greater emphasis on the value of follow-up in our series and in the management of soft tissue sarcomas in general, particularly with regard to the early detection of local recurrences. To support this, we have added a new recent bibliographic reference that reviews the topic in the final paragraph on the limitations and strengths of the study:

Lines 434-437: In this regard, the average follow-up period for our series was 39 months, with 6 patients having more than 3 years of follow-up and 3 of them having more than 6 years. In any case, an active surveillance approach is essential for the early detection of local recurrences [42], especially in high-grade LMS.

Diagnostic Markers: It would be beneficial to describe possible diagnostic markers that help reduce unplanned excisions, thereby reinforcing the clinical impact of the study.

The specific paragraph referring to this issue has been completed and two relevant bibliographical references have been added:

Lines 410-415: As established in the guidelines, any soft tissue mass deeper than the fascia and larger than 5 cm in diameter, whether painful or not, especially if it grows or reappears after resection, is a sarcoma until proven otherwise [39]. The same axiom should apply to subcutaneous tumours, bearing in mind that one third of soft tissue sarcomas are superficial and that their size is usually smaller than that of deep SPBs [40].

In-depth Discussion of Neuropathic Pain: Although the authors mention the presence and management of neuropathic pain, it is suggested to expand this discussion. They could address how neuropathic pain affects the patient's quality of life, detail management strategies, and highlight the need for future research on this complication in this type of tumor.

This complication has been added to the Discussion section where its possible causes are analyzed. We highlight the need to consider this issue in light of the close anatomical relationship between the saphenous nerve and the great saphenous vein: 

Lines 371-375: The neuropathic pain reported by case 3 was attributed to a surgical complication directly related to the resection of terminal branches of the saphenous nerve and the use of a graft for soft tissue coverage. This eventuality, which is sometimes challenging to treat, should be taken into consideration due to the proximity of the nerve to the GSV.

Minor changes have been made on the quality of English language of the manuscript in an attempt to improve its readability. More particularly, we have revised several paragraphs within the Abstract, Introduction, Material and Methods, and Conclusion sections, while ensuring that the content remains consistent with the original version of the manuscript.

All changes suggested by the reviewers -many of which are summarised and confirmed in the tables and figures of the manuscript- have been taken into account. We sincerely appreciate the comments, which we believe enrich our paper.

Round 2

Reviewer 1 Report

Comments and Suggestions for Authors

No more comments

Author Response

Comments: No more comments.

Response: Thank you

Reviewer 2 Report

Comments and Suggestions for Authors

The manuscript has been well-revised.

It would be better to describe histological subtypes (variants) of 9 cases, if possible.

It would be better to revise the title of the paper as follows:

Leiomyosarcomas of the Great Saphenous Vein. Diagnostic Management to Avoid Unplanned Excisions and Oncologic, Functional and Psychological Treatment Results

-> Leiomyosarcomas of the Great Saphenous Vein: Diagnostic and Therapeutic Strategies to Prevent Unplanned Excisions and Improve Oncologic, Functional, and Psychological Outcomes

Comments on the Quality of English Language

Please check English grammar and spelling.
For example,
 Line 278: Subcutaneous LMS account -> Subcutaneous  LMS accounts
 Line 322: strogen -> estrogen

Author Response

Comment 1: It would be better to describe histological subtypes (variants) of 9 cases, if possible.

Response 1:  Certainly, although LMS subtypes share uniform histologic features, their clinical behavior and characteristics differ markedly according to site of origin (uterus, retroperitoneum, vena cava, subcutaneous or cutaneous tissues, etc.). In our series, histologic subtyping was not feasible beyond classification as subcutaneous LMS, and we therefore assumed a comparable clinical course for tumors of the same histologic grade. Recent studies emphasize the need for molecular analyses and prospective investigations to assess the efficacy of emerging immunotherapeutic strategies. We have accordingly acknowledged this as a limitation of our study and cited a 2025 reference addressing this issue: Lagos G, Groisberg R, Elliott A, Dizon DS, Seeber A, Gibney GT, von Mehren M, Cardona K, Demeure MJ, Riedel RF, Florou V, Chou AJ, Modiano JF, Kumar A, Khushman MM, D'Amato GZ, Espejo Freire AP, DeNardo B, Trent JC. Large-Scale Multiomic Analysis Identifies Anatomic Differences and Immunogenic Potential in Subtypes of Leiomyosarcoma. Clin Cancer Res. 2025 Jun 3;31(11):2210-2221. doi: 10.1158/1078-0432.CCR-24-2503. PMID: 40100098.

“The third limitation is that we did not classify our cases into histologic subtypes of LMS, but rather considered them collectively as subcutaneous leiomyosarcomas of varying malignant grade. Although LMS subtypes share uniform histologic features, their clinical behavior and characteristics differ substantially across anatomic sites (uterus, retroperitoneum, vena cava, subcutaneous or cutaneous tissues, etc.), and molecular studies appear necessary to assess the efficacy of novel immunotherapeutic approaches [42]”

Comment 2: It would be better to revise the title of the paper as follows:

Leiomyosarcomas of the Great Saphenous Vein. Diagnostic Management to Avoid Unplanned Excisions and Oncologic, Functional and Psychological Treatment Results
-> Leiomyosarcomas of the Great Saphenous Vein: Diagnostic and Therapeutic Strategies to Prevent Unplanned Excisions and Improve Oncologic, Functional, and Psychological Outcomes

Response 2: The manuscript title has been modified as suggested by the reviewer.

Comments on the Quality of English Language

Comments 3: Please check English grammar and spelling. For example,
 Line 278: Subcutaneous LMS account -> Subcutaneous  LMS accounts
 Line 322: strogen -> estrogen

Response 3: Both typographical errors have been corrected.

Reviewer 3 Report

Comments and Suggestions for Authors

Thank you for the opportunity to review the revised version of the manuscript. The authors have introduced revisions in the text and changes in the English language in order to improve the readability, and regarding this aspect the manuscript is improved. The central point of criticism of course remains, there is to my opinion no significant novelty and the results come from a very small patient cohort, adding nothing important to what we already know about this type of tumours. 

Author Response

Comments: Thank you for the opportunity to review the revised version of the manuscript. The authors have introduced revisions in the text and changes in the English language in order to improve the readability, and regarding this aspect the manuscript is improved. The central point of criticism of course remains, there is to my opinion no significant novelty and the results come from a very small patient cohort, adding nothing important to what we already know about this type of tumours. 

Response: We greatly appreciate your insightful comments, which have contributed to improving the manuscript, as well as your recognition of our work. With regard to your critique, it is true that unplanned resections remain a recurrent topic in the scientific literature; however, the problem persists in daily clinical practice. For this reason, and because, to our knowledge, this is the largest published series of leiomyosarcomas of the great saphenous vein treated by a single multidisciplinary team worldwide, we believe our study holds both practical and epidemiological relevance.

Reviewer 4 Report

Comments and Suggestions for Authors

The manuscript has been improved

Author Response

Comments: The manuscript has been improved

Response: Thank you

Round 3

Reviewer 3 Report

Comments and Suggestions for Authors

Thank you for the revised version. I respect the authors reply. Some minor changes have further improved the manuscript, which is a well-written paper but in essence suffering from the main shortcoming which to my opinion is central: It is a subclassification, based on the anatomical location, of a known diagnosis with only  9 patients. 

To my opinion, for the paper to be publishable, a metanalysis of all the published patients in the last regarding overall survival and local control of the disease should be done. I Believe the inclusion criteria should be longer than 20 years, as set in the present paper, in PMID: 30500647 i noticed that many cases were publised in the 1990s and surgical treatment, radiotherapy and prognosis have not changed considerably since then, so thay may be included. 

Author Response

Comments 1:

Thank you for the revised version. I respect the authors reply. Some minor changes have further improved the manuscript, which is a well-written paper but in essence suffering from the main shortcoming which to my opinion is central: It is a subclassification, based on the anatomical location, of a known diagnosis with only  9 patients. 

To my opinion, for the paper to be publishable, a metanalysis of all the published patients in the last regarding overall survival and local control of the disease should be done. I Believe the inclusion criteria should be longer than 20 years, as set in the present paper, in PMID: 30500647 i noticed that many cases were publised in the 1990s and surgical treatment, radiotherapy and prognosis have not changed considerably since then, so thay may be included. 

Response 1:

A literature review was conducted using the PubMed database. The search strategy included the terms “leiomyosarcoma” AND “saphenous” AND “vein” AND “great”/“greater”, with no time restrictions applied. The search retrieved 33 records. Among these, 7 articles were excluded as they did not address the topic of interest. Of the remaining 26, 15 had already been cited in the previous version of our manuscript and were therefore excluded from further analysis. Finally, 11 new articles were included, together with one indirectly referenced case (Berlin et al., 1984). Each of these articles described a single case of GSV-LMS, which confirms that our series represents the largest published to date in the scientific literature.

In addition, 14 new articles providing relevant information on overall survival and local disease control were incorporated into Table 3.

All the new articles included were added to the reference list, and the numbering of citations was updated accordingly. The final paragraph of the Introduction was also revised to integrate the updated epidemiological data, as follows: “As of 2023, four literature reviews had identified a total of 48 cases reported across 45 publications [3–6]. In the latest literature review conducted in 2025, compiling cases from 1868 to 2024, a total of 56 cases in 53 reports were identified [7]. To this number, one additional unreported case involving the thigh should be added [8]. Among the 61 total papers, 57 presented a solitary case [3-29] and 3 presented two cases each [30-32]”.

Thank you for your comments